# Spatiotemporal Variations in Agricultural Flooding in Middle and Lower Reaches of Yangtze River from 1970 to 2018

**Shuqi Wu [1], Shisong Cao [2,\*], Zhiheng Wang [3], Xinyuan Qu [1], Shanfei Li [4] and Wenji Zhao [1,\*]**

1   School of Resource, Environment and Tourism, Capital Normal University, Beijing 100048, China;
    2180902151@cnu.edu.cn (S.W.); qqzwind@gmail.com (X.Q.)
2   School of Surveying and Urban Space Information, Beijing University of Civil Engineering and Architecture,
    Beijing 100044, China
3   School of Geology and Geomatics, Tianjin Chengjian University, Tianjin 300384, China;
    wangzhiheng@tcu.edu.cn
4   Beijing Engineering Corporation Limited, Beijing 100024, China; lishanfei1990@gmail.com
\*   Correspondence: caoshisong@bucea.edu.cn (S.C.); 4973@cnu.edu.cn (W.Z.);
    Tel.: +86-18612970387 (S.C.); +86-13810707928 (W.Z.)

**Abstract:** Agricultural floods in the middle and lower reaches of the Yangtze River, known as the "land of fish and rice" in China, have increased both in areal coverage and frequency over the past 50 years, presenting a grave challenge to sustainable development and food security in the region. Studying the spatiotemporal variation characteristics of agricultural floods in this region is thus important for providing a scientific basis for regional flood control and disaster mitigation. We used variation trend analyses, Mann–Kendall tests, wavelet analyses, and center of gravity modeling to study spatiotemporal changes in agricultural floods in the study area, based on agricultural flood indicators. Changes in agricultural flood frequency showed an overall increasing trend. The frequency of floods changed abruptly in 1990, with the average frequency of floods per station increasing by 0.2086/year from 1991 through 2018, characterized by multiple time-scale changes. The time scale of 17 years had three low–high cycles, that of eight years had six, and that of four years had 13. Agricultural floods in the study area were concentrated in the southern Yangtze River and mainly occurred in northeastern Jiangxi Province and the southeastern Zhejiang Province. The area with high agricultural flood indices increased. Agricultural floods were closely related to the Yangtze River and the direction of the gravity center of agricultural floods was similar to that of the river. Affected by precipitation intensity and frequency, the gravity center fluctuated greatly and generally alternated from southwest to northeast.

**Keywords:** agricultural flood; center of gravity modeling; flood-level indicator; Mann–Kendall; wavelet analysis

## 1. Introduction

Extreme weather events and floods have occurred more frequently since the 1960s, bringing about various environmental problems and causing huge economic losses and severe challenges to sustainable development [1,2]. China has experienced some of the most serious flood-related losses in terms of its economy and population [3]. As a large agricultural country, China uses 7% of its cultivated land to support 22% of its population [4]. Over the next 20 years, the nation's food requirements are expected to increase by 30–50% [5]. The flat terrain of the middle and lower reaches of the Yangtze River, which supports China's major agricultural production areas, experiences a monsoon climate

that often brings about flooding. The monsoon cycle, increased spatiotemporal variations of rainfall, the population density, and concentrated areas of high economic activity have increased the risk of disasters in the region [6]. The agricultural sector is particularly vulnerable to economic losses caused by floods [7], and frequent floods therefore affect the development of the local agricultural economy. Understanding the spatiotemporal variation characteristics of agricultural flood disasters is necessary for good scientific understanding of the middle and lower reaches of the Yangtze River in order to achieve regional flood control and disaster mitigation.

Current research on the impact of flood disasters can be roughly divided into two time-scale categories, as follows: Small time scales, which focus on critical rainfall in rainstorm flooding [8,9], and large spatiotemporal scales, which use meteorological, historical, or disaster data to conduct risk zoning and spatiotemporal distribution research. The main research methods of flood risk zoning include using remote sensing technology to monitor and evaluate flood disasters [10,11], using historical disaster data to assess flood disaster risks [12,13], and the use of hydrodynamic models for scenario simulation [14,15]. Research on spatiotemporal changes in flood disasters mainly involves selecting flood indicators and flood-level classification. Commonly used flood indicators include peak flow, water level [16], Z-index [17], annual average rainfall [18], and seasonal rainfall [19]. Flood-level research methods include the classification of flood levels by flow [20,21] and the percentile threshold method [22,23]. Llasat et al. [24] studied the evolution of floods in the northwestern Mediterranean and divided floods into three levels—general, extraordinarily large, and destructive, according to the severity of the resulting disaster. Wan et al. [25] used the disaster data of provincial economic losses, disaster-affected areas, the deceased and injured population, and direct economic losses to classify flood levels. Chen et al. [26] used fuzzy clustering analysis to analyze the flood disasters of 30 provinces in China in 2008 and showed that this method is suitable for flood disaster classification.

Research on flood disasters in the middle and lower reaches of the Yangtze River has mainly focused on precipitation characteristics, the causes of storm floods, and flood disaster risk analysis [27–29]. However, few reports exist on agricultural flood indicators and their spatiotemporal distribution characteristics in this region. In this paper, we used meteorological data combined with actual agricultural flood data to construct indicators, and we tested the practicality of indicators in agricultural flood monitoring and early warning assessment. The frequency, trend, and spatial differences in agricultural flood events in the region were studied to provide a reference for disaster prevention, mitigation, and early warnings based on the correlation between agricultural floods and rainstorm flooding.

## 2. Study Area

The middle and lower reaches of the Yangtze River (106°5′–121°54′ and 24°29′–34°11′) contain the Dongting Lake, Poyang Lake, Han River, and Taihu Basins (Figure 1). The area has a north subtropical monsoon climate, with more than 50% of the precipitation occurring in June–August, and the annual variation in precipitation is large. The study area is an important agricultural development zone in China, producing grain, oil, and cotton, with a food commodity rate of over 40% [30]. However, flood disasters have become a serious obstacle to the sustainable development of agriculture in the region. Between 1206 and 1949, 1092 rainstorm floods have occurred [29]. The 2016 flood in the Yangtze River Basin afflicted a total of 456.114 million ha of crops, inundating an area of 271.513 million ha, with 102.191 million ha yielding no crops, and causing an economic loss of 1.704277 trillion yuan.

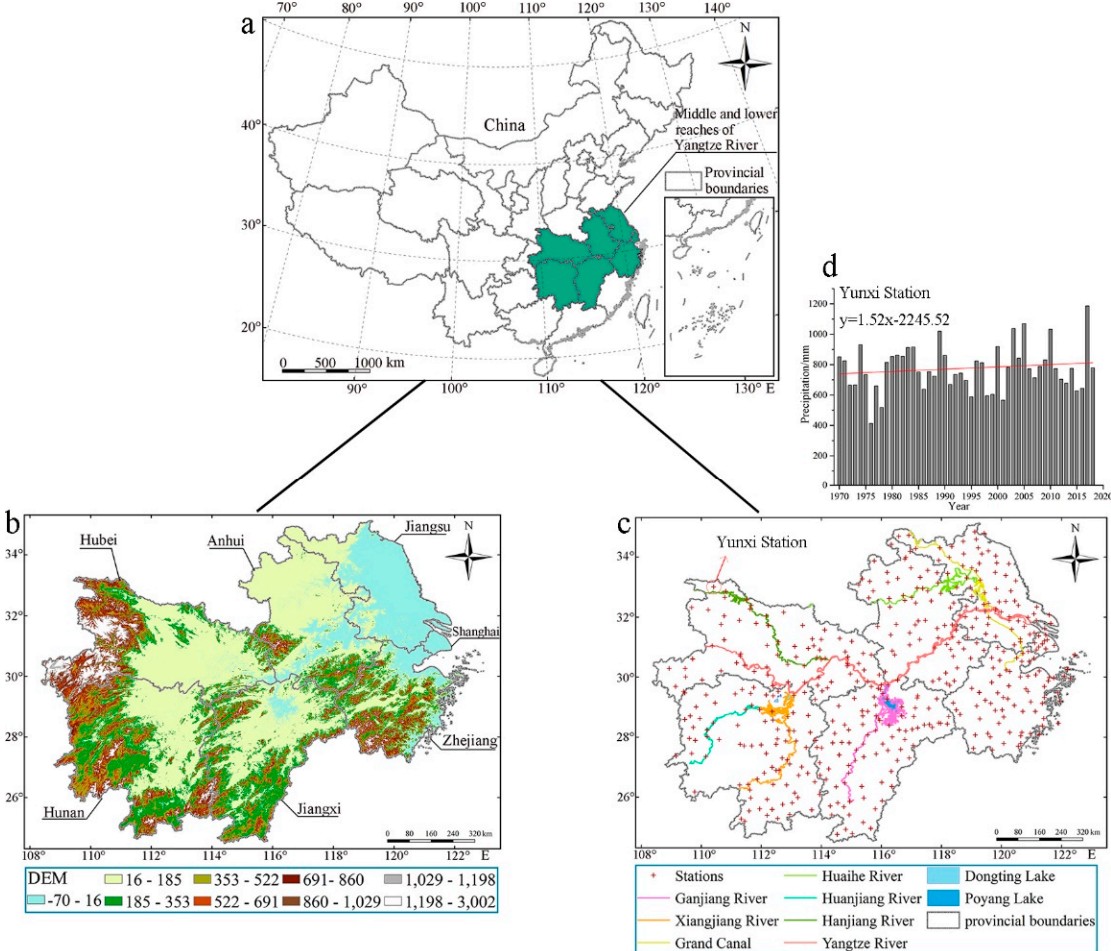

**Figure 1.** (**a**) Location, provincial boundaries; (**b**) Terrain; (**c**) Rivers and distribution of meteorological stations in middle and lower reaches of Yangtze River, China. (**d**) An example of precipitation. DEM: digital elevation model.

## 3. Data Sources and Research Methods

### 3.1. Data Sources

Meteorological data used in this study were obtained from the China Meteorological Administration, including daily precipitation data of 489 ground monitoring stations in Zhejiang, Jiangsu, Shanghai, Anhui, Hunan, and Hubei Provinces from 1970 to 2018. Stations with missing data were omitted, and data from 435 stations were retained. Agricultural flood data were derived from the statistical yearbooks of the provinces and cities in the study, including the spatial extent of flooding, damaged, and inundated areas. Agricultural floods were characterized by the flood disaster rate over the study period, which reflected the proportion of the inundated area. Historical flood data, including the flood depth, duration, and disaster losses during floods in different provinces from 1970 to 2018, were mainly obtained from meteorological disaster books and records. This study utilized the middle and lower reaches of the Yangtze River as the main study area, and the counties and cities under its jurisdiction were used as subregions (Table 1).

**Table 1.** Statistical data used in this paper.

| Data Type | Data Sources |
| --- | --- |
| Planting area (ha) | Statistical yearbooks |
| Affected people (population) | Statistical yearbooks |
| Damage area (ha) | Statistical yearbooks |
| Economic loss (trillion yuan) | Meteorological disaster books |
| Flood depth (m) | Meteorological disaster books |
| Flood duration (d) | Meteorological disaster books |

*3.2. Research Methods*

3.2.1. Agricultural Flood Indicators

Flood Classification

Agricultural flood indicators can generally be constructed using correlation analyses of rainstorm flood indicators and the proportion of the inundated area [31]. Rainfall is a direct cause of and trigger for flooding. Continuous extreme precipitation events often lead to floods and have been extensively researched [32]. In most cases, floods are caused by a high rainfall event over a short period of time (usually 1–3 days). Flooding often occurs during continuous precipitation, e.g., over one day. The process of cumulative precipitation refers to continuous precipitation of more than 50 mm over one day. When less than 0.1 mm of daily precipitation occurs over multiple days, the process of cumulative precipitation is considered interrupted. According to the "Rainstorm Flooding Level Standard", this study used the division of rainstorm flooding levels outlined by Zhang [33] and Wan [34]. Precipitation data from 1970 to 2018 were used to determine the cumulative frequency of floods throughout the middle and lower reaches of the Yangtze River (Table 2).

**Table 2.** The level classification standards for floods used in this study.

| Continuous Rainy Days (d) | Total Rainfall, R (mm) | Flood Level |
| --- | --- | --- |
| 1 | $80 \leq R < 100$ | Waterlogging |
| | $100 \leq R < 150$ | Medium flooding |
| | $150 \leq R < 200$ | Severe flooding |
| | $R \geq 200$ | Catastrophic flooding |
| 2 | $100 \leq R < 150$ | Waterlogging |
| | $150 \leq R < 200$ | Medium flooding |
| | $200 \leq R < 250$ | Severe flooding |
| | $R \geq 250$ | Catastrophic flooding |
| 3 | $150 \leq R < 200$ | Waterlogging |
| | $200 \leq R < 250$ | Medium flooding |
| | $250 \leq R < 300$ | Severe flooding |
| | $R \geq 300$ | Catastrophic flooding |

Multiple Linear Regression

The agricultural flood data obtained in this study included the provincial agricultural flood damage area from 1970 to 2018. To match the number of floods at a single station with the disaster data and to evaluate the total flood intensity of a single station, we defined the single-station annual agricultural flood and provincial annual agricultural flood indices as follows:

$$p = w1 \times f1 + w2 \times f2 + w3 \times f3 + w4 \times f4, \tag{1}$$

$$P = \frac{w1 \times F1 + w2 \times F2 + w3 \times F3 + w4 \times F4}{N}, \tag{2}$$

where $p$ represents the single-station annual agricultural flood indices; $f1$, $f2$, $f3$, and $f4$ are the frequencies of waterlogging, medium flooding, severe flooding, and catastrophic flooding, respectively, at a single station in one year; $F1$, $F2$, $F3$, and $F4$ are the sum of the frequencies of waterlogging, medium flooding, severe flooding, and catastrophic flooding, respectively, at all stations in a province in one year; $P$ represents the provincial annual agricultural flood indices; $N$ is the total number of stations in the province; and $w1$, $w2$, $w3$, and $w4$ are the weights of the frequencies of waterlogging, medium flooding, severe flooding, and catastrophic flooding, respectively.

Multivariate linear regressions were used to determine the weight of the frequency of waterlogging, medium flooding, severe flooding, and catastrophic flooding. We used sequence $Y1$ of each province's annual flood disaster rate and frequency sequences $N1$, $N2$, $N3$, and $N4$ of each province's waterlogging, medium flooding, severe flooding, and catastrophic flooding values to establish a multivariate linear regression equation. It must be noted that the annual rate of disasters in each province was linked to the frequency of floods in that province.

$$Y_1 = \beta_0 + \beta_1 F_1 + \beta_2 F_2 + \beta_3 F_3 + \beta_4 F_4, \tag{3}$$

where $\beta_1$, $\beta_2$, $\beta_3$, and $\beta_4$ are all greater than 0, and the regression equation satisfies the significance test. We eliminated indicators of less than 0 in the multiple linear regression equation, as follows:

$$w_i = \frac{\beta_i}{\sum\limits_{i=1}^{4} \beta_i}, i = 1, 2, 3, 4 \tag{4}$$

In addition, to eliminate the impact of the number of sites in each province, the average annual agricultural flood frequency per station was defined as follows:

$$\overline{F}_i = \sum_{j=1}^{n} F_{ij}/N, \tag{5}$$

where $\overline{F}_i$ is the annual average agricultural flood frequency per station in the province; $N$ is the number of meteorological stations in each province and city; $i$ is defined as 1, 2, 3, and 4 to represent waterlogging, medium flooding, severe flooding, and catastrophic flooding, respectively; and $F_{ij}$ is the annual flood frequency of each level of flooding.

### 3.2.2. Analysis of Time Series Variation

#### Mann–Kendall Abrupt Change Detection

For abrupt climate change detection, the Mann–Kendall nonparametric detection method has the characteristics of a wide detection range, less human interference, and a high quantitative degree. The sample does not have to comply with a certain distribution, and is not subject to interference by a few outliers. It has been widely used to detect trends in hydrological and meteorological time series [35,36]. The null hypothesis, $H_0$, was as follows: If the climate sequence has not changed, it is $x_1 \, x_2 \ldots x_n$, and $m_i$ indicates the cumulative number of the $i$th sample when $x_i > x_j (1 \le j \le i)$. Subsequently, $d_k$ is defined as follows:

$$d_k = \sum_{i=1}^{k} m_i (2 \le k \le n), \tag{6}$$

and $d_k$ can be normalized as follows:

$$u(d_k) = \frac{d_k - E[d_k]}{\sqrt{\text{var}[d_k]}}. \tag{7}$$

The probability $\alpha = P(|u| > |u(d_k)|)$ and $u(d_k)$ meets the standard normal distribution and can be obtained via calculation or from table, as follows:

$$E(d_k) = \frac{k(k-1)}{4}, \tag{8}$$

and

$$\text{var}(d_k) = \frac{k(k-1)(2k+5)}{72}(2 \le k \le n). \tag{9}$$

Equations (8) and (9) represent the mathematical expectation and variance of $d_k$, respectively. Given the significance level $\alpha_0$, when $\alpha > \alpha_0$, the null hypothesis, $H_0$, was accepted. When $\alpha < \alpha_0$, the null hypothesis was negated, indicating that there was a significant increase or decrease in the sample sequence. In addition, $u(d_k)(1 \le k \le n)$ forms a curve, *UF*, through the confidence test to determine its trend. This method was applied to the inverse sequence, where $\overline{m}_i$ is the cumulative number of the $i$th sample when $x_i > x_j(i \le j \le n)$. When $i' = n+1-i$, if $\overline{m} = m_i$, the inverse sequence $\overline{u} = (d_i)$ was obtained as follows:

$$\overline{u}(d_k) = -u(d_i),\ i' = n+1-i,\ (i, i' = 1, 2, \dots, n). \tag{10}$$

Consequently, $\overline{U}(d_k)$ was drawn as curve *UB*. When curve *UF* exceeded the reliability line, there was a clear trend. If the intersection of *UF* and *UB* was between the confidence lines, this point represented the beginning of the mutation point.

Wavelet Periodic Analysis

The cumulative frequency of floods is presented in Table 2. Due to the influence of natural factors, the time series of flood frequency, has complex, nonlinear, and multi-time-scale variation characteristics. Orthogonal or discrete wavelet transform cannot accurately analyze the signal. Compared with real wavelet transform, complex wavelet transform can not only display the amplitude of the sequence change, but can also provide the phase of the time series change, rendering the signal analysis deeper. In addition, complex wavelet transform can avoid errors caused by false oscillations and can improve the accuracy of the results. This method has been widely applied to the study of multi-scale climate-change characteristics. Therefore, continuous Morlet complex wavelet transform was used to analyze the multi-time characteristics of flood frequency [37,38]:

When the number of consecutive rainy days exceeded 3 days, the total rainfall was calculated by taking the three days with the highest rainfall.

$$\int \psi(t) = 0, \tag{11}$$

where $\psi(t)$ is a basic wavelet function, using the following wavelet transform:

$$W_f(a, b) = |a|^{-\frac{1}{2}} \int_R f(t)\overline{\psi}(\frac{t-b}{a})dt, \tag{12}$$

where $W_f(a, b)$ is the wavelet transform coefficient; $f(t)$ is a signal or square integrable function; $a$ is the extension scale; $b$ is the translation parameter; and $\overline{\psi}(\frac{t-b}{a})$ is the complex conjugate function of $\overline{\psi}(\frac{a-b}{a})$. In geosciences, most time series data are discrete. If the function $f(t) = f(k\Delta t)$ ($k = 1, 2 \dots N$; $\Delta t$ is the time interval), the discrete wavelet can be transformed as follows:

$$W_f(a,b) = |a|^{-\frac{1}{2}} \Delta t \sum_{k=1}^{N} f(k\Delta t) \overline{\psi}(\frac{k\Delta t - b}{a}) dt. \tag{13}$$

By integrating all the wavelet coefficients of different time scales obtained through the wavelet transform equation on the *b* domain, the wavelet variance could be obtained, as follows:

$$Var(a) = \int_{-\infty}^{\infty} \left| W_f(a,b) \right|^2 db. \tag{14}$$

The wavelet variance diagram shows the variation in wavelet variance on time scale *a*, which can reflect the energy distribution of the signal fluctuation with scale *a*. Therefore, the wavelet variance diagram can detect the main time scale in the signal, which is the main period.

### 3.2.3. Spatial Change Analysis

Trend Analysis

Based on the trend in the agricultural flood index of the pixel, the slope of the annual average agricultural flood index was fitted using the least squares method, which comprehensively reflects the spatiotemporal pattern evolution of the regional agricultural flood index, as follows:

$$\theta_{\text{slope}} = \frac{n\sum\limits_{i=1}^{n} iR_i - \sum\limits_{i=1}^{n} i \sum\limits_{i=1}^{n} R_i}{n\sum\limits_{i=1}^{n} i^2 - (\sum\limits_{i=1}^{n} i)^2}, \tag{15}$$

where $\theta_{\text{slope}}$ is the trend of the slope; *n* is the monitoring period (years); and $R_i$ is the agricultural flood index of year *i*. Positive or negative slopes respectively indicate an increased or decreased agricultural flood index. Significance was determined using F tests and represents the trend credibility level, but is not related to the rate of change, as follows:

$$F = U \times \frac{N-2}{Q}, \tag{16}$$

where $U = \sum\limits_{i=1}^{n} (\hat{y}_i - \overline{y})^2$ is the sum of squares; $Q = \sum\limits_{i=1}^{n} (y_i - \hat{y})^2$ is the regression sum of squares; $y_i$ is the agricultural flood index in year *i*; $\hat{y}_i$ is the average value of the agricultural flood index for 49 years; and *n* is the number of years (49). Variation trends could be divided into the following six levels: Highly significantly reduced ($\theta_{\text{slope}} < 0$, $p < 0.01$), significantly reduced ($\theta_{\text{slope}} < 0$, $p < 0.05$), non-significantly reduced ($\theta_{\text{slope}} < 0$, $p > 0.05$), non-significantly increased ($\theta_{\text{slope}} > 0$, $p > 0.05$), significantly increased ($\theta_{\text{slope}} > 0$, $p < 0.05$), and highly significantly increased ($\theta_{\text{slope}} > 0$, $p < 0.01$).

Center of Gravity Transfer Analysis

For the center of gravity and its moving direction, moving distance can not only indicate the regional geographical phenomenon spatial difference, but can also show its dynamic processes and evolutionary law. This study used the gravity center migration model to quantitatively study the spatial variation in agricultural floods in the middle and lower reaches of the Yangtze River, as follows:

$$X_t = \sum_{i=1}^{n} x_i M_i / \sum_{i=1}^{n} M_i \tag{17}$$

$$Y_t = \sum_{i=1}^{n} y_i M_i \bigg/ \sum_{i=1}^{n} M_i \qquad (18)$$

where $X_t$ and $Y_t$ respectively represent the latitude and longitude of the gravity center of the agricultural flood index in year $t$; $x_i$ and $y_i$ represent the latitude and longitude of the gravity center of the $i$th subregion, respectively [39]; and $M_i$ represents the measure of a certain attribute of the $i$th subregion. The latitude and longitude of the administrative center each county and city were taken as calculation parameters, and the county-level agricultural flood indices were used as the calculated attribute value. The indices were dimensionless to ensure the feasibility of the data.

The displacement distance of the gravity center of the area was generally expressed as follows:

$$D_{t_2-t_1} = C \sqrt{(X_{t_2} - X_{t_1})^2 + (Y_{t_2} - Y_{t_1})^2} \qquad (19)$$

where $D_{t_2-t_1}$ indicates the moving distance of the center of gravity between two years; $t_1$ and $t_2$ represent different years; $(X_{t_1} Y_{t_1})$ and $(X_{t_2} Y_{t_2})$ represent the geographic coordinates of the center of gravity of years $t_1$ and $t_2$; and $C$ is a constant (111.111) and is a coefficient that converts geographic coordinates (latitude and longitude) into plane distance (km). Equation (20) was used to determine the spatial movement direction of the gravity center of the region, as follows:

$$\theta_{t_2-t_1} = \frac{n\pi}{2} + arctg\left(\frac{y_{t_2} - y_{t_1}}{x_{t_2} - x_{t_1}}\right) \qquad (20)$$

where $\theta_{t_2-t_1}$ indicates the angle at which the center of gravity of the agricultural floods moves from year $t_1$ to $t_2$. The eastern direction was taken as 0°, and clockwise rotation was negative and counterclockwise rotation was positive. When the center of gravity moves northeastwardly, $0° < \theta < 90°$; when it moves northwestwardly, $90° < \theta < 180°$; when it moves southeastwardly, $-90° < \theta < 0°$; and when it moves southwestwardly, $-180° < \theta < -90°$.

## 4. Process and Result Analysis

### 4.1. Analysis of Temporal Variation Characteristics of Agricultural Flooding

4.1.1. Analysis of Trends in Agricultural Flooding

Regression and anomaly analyses were used to reveal the characteristics and laws of the overall changes in agricultural flooding in the study area from 1970 to 2018. To reduce the impact of extreme and abnormal events on long-term trend analyses, we used a 5-year moving average analysis on the anomalies. Combined with the Mann–Kendall test and cumulative anomaly methods, the abrupt changes in annual flood frequency in the region were tested at a significance level of 0.05 (95% confidence detection line). Finally, we obtained the graphs of the trend in the flood frequency per station over time, the anomaly, Mann–Kendall abrupt change analysis, and the cumulative anomaly (Figure 2).

Overall, the agricultural flood frequency in the study area showed an increasing trend, and the trend line slope was 0.007/year. Over the past 49 years, the anomaly of the agricultural flood frequency fluctuated greatly, ranging from −0.4 to 0.7; the minimum appeared in 1985, and the maximum appeared in 2016. The positive anomaly accumulative year was 22 years, and that of negative anomaly was 27 years. The 5-year moving average indicated that the annual flood frequency per station in the region experienced "growth–decrease–growth–decrease–growth" fluctuations. Three declines occurred in the overall increasing trend, namely from 1974 to 1980, 1982 to 1986, and 1997 to 2001. The abrupt change in the annual flood frequency per station shows that the UF curve experienced two fluctuations between 1970 and 1985, after which it continued to increase. The UF and UB curves intersected twice between 1990 and 1995, and the intersections were within the confidence interval. The UF curve exceeded the confidence level around 1996. The cumulative anomaly continued to

decline between 1970 and 1990, with a fluctuation between 1980 and 1985. It showed an increasing trend between 1990 and 2018, was relatively flat between 2000 and 2010, and increased rapidly after 2010. Based on Mann–Kendall mutation analyses and cumulative anomaly changes, the agricultural flood frequency abrupt change was 1990. Between 1970 and 1990, the average agricultural flood frequency per station showed a decreasing trend, and the trend line slope was −0.0005/year. From 1991 to 2018, the average agricultural flood frequency per station showed an increasing trend, with a slope of 0.0017/year. The average agricultural flood frequency per station increased by 0.2086/year in 1991–2018.

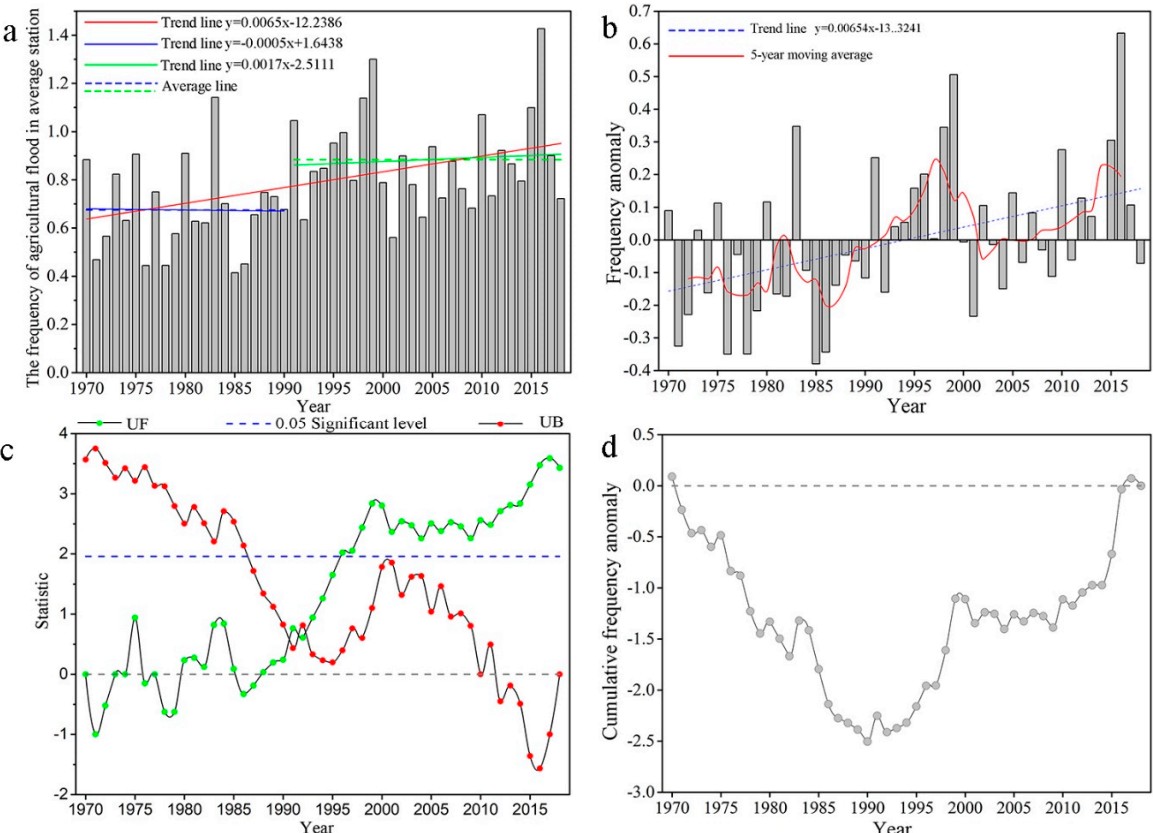

**Figure 2.** (**a**) Trend, (**b**) anomaly, (**c**) Mann–Kendall test, and (**d**) cumulative anomaly of agricultural flood frequency in the middle and lower reaches of the Yangtze River, China, from 1970 to 2018.

### 4.1.2. Analysis of Period Change in Agricultural Flood

Figure 3 shows the real-part contour plot of the wavelet coefficients and a variance plot. When the real part of the wavelet coefficient was positive, the annual agricultural flood frequency increased, and when it was negative, the annual agricultural flood frequency decreased. The wavelet variance map shows the main period in the development of the annual flood frequency. Each time scale of the annual agricultural flood frequency sequence in the time domain was unevenly distributed, and the local variation characteristics were obvious. In general, the annual agricultural flood frequency had three time scales; namely, 2–4, 5–10, and 13–30 years. Three distinct peaks occurred in the wavelet variance, which corresponded to the time scales of 4, 8, and 17 years, respectively. The 17-year time scale corresponded to the maximum peak, indicating that a periodic oscillation of 17 years was the strongest, which was considered the first main period of agricultural flood frequency change. The 8-year time scale was the second main period, and the 4-year time scale was the third.

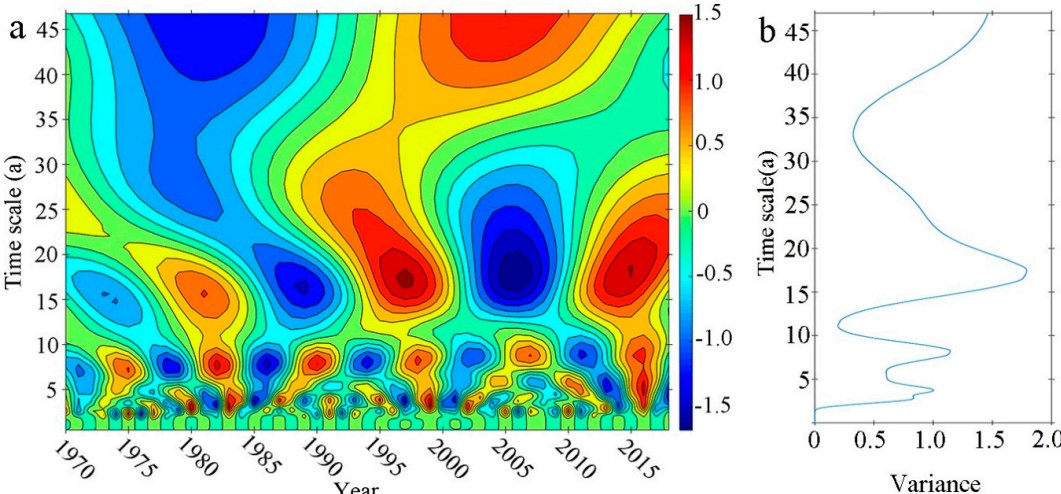

**Figure 3.** (**a**) Real-part wavelet transform and (**b**) wavelet variance of annual agricultural flood frequency.

Figure 4 shows the variation in the main period wavelet coefficients. We analyzed the high and low changes in annual agricultural flood frequency at different time scales. In the 17-year time scale, the annual agricultural flood frequency had three cycles from low to high, in the 8-year time scale it had six cycles, and in the 4-year time scale it had 13 cycles. In addition, the high–low variation of the large time scale of the annual flood frequency included the high–low variation of the smaller time scale, for example, the higher period of the first agricultural flood frequency at the 17-year scale included one high and one low period on the 8-year time scale and two low and one high periods on the 4-year time scale, and the lower period of the second agricultural flood frequency at the 17-year scale included one high and one low period on the 8-year time scale and two low and two high periods on the 4-year time scale. The recent trend in agricultural flood frequency in the study area, according to the wavelet coefficient values of the 17- and 8-year periods, indicated a peak in the annual agricultural flood frequency around 2015. Furthermore, the frequency of agricultural floods was estimated to decrease after 2018. According to the wavelet coefficient value of the 4-year time scale, the minimum value was reached around 2018, and the agricultural flood frequency was estimated to increase after 2018.

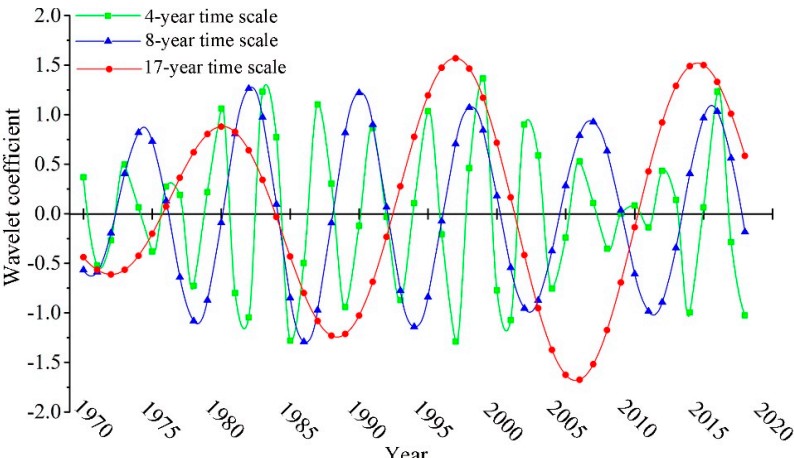

**Figure 4.** Changes in wavelet coefficients of annual agricultural flood frequency.

*4.2. Analysis of Spatial Variation Characteristics of Agricultural Flooding*

4.2.1. Spatial Distribution Characteristics of Agricultural Flooding

Figure 5 shows the average distribution of the frequency of the agricultural flood indices in the middle and lower reaches of the Yangtze River from 1970 to 2018. The flood index reflects the overall

situation of agricultural flooding. Many floods occurred in the southern Yellow River, but few occurred in the north. Frequent flooding mainly occurred in northeastern Jiangxi province, the junction of Jiangxi, Anhui, and Zhejiang Provinces, Poyang Lake, Dongting Lake, Zhangjiajie City, Enshi City, and Jianghan Plain, and in southeastern Zhejiang Province. Flood disasters were particularly serious in Zhejiang Province from 1949 to 1998, especially in the hills and plains of Ningbo, Taizhou, and Wenzhou Cities on the southeastern coast [40]. This may be related to the long-term impact of typhoons landing in the coastal areas of the province [41]. In addition, these areas that frequently experience flooding are mainly located in low-lying lands and around lakes or rivers, which are prone to flooding disasters. In turn, the areas that experience waterlogging and medium, severe, and catastrophic floods have decreased. In particular, the junction of Jiangxi, Anhui, and Zhejiang Provinces and the eastern coastal areas of Zhejiang Province frequently experience flooding. Northwestern Hubei Province, northern Anhui Province, and northern Jiangsu Province are low-incidence areas for floods.

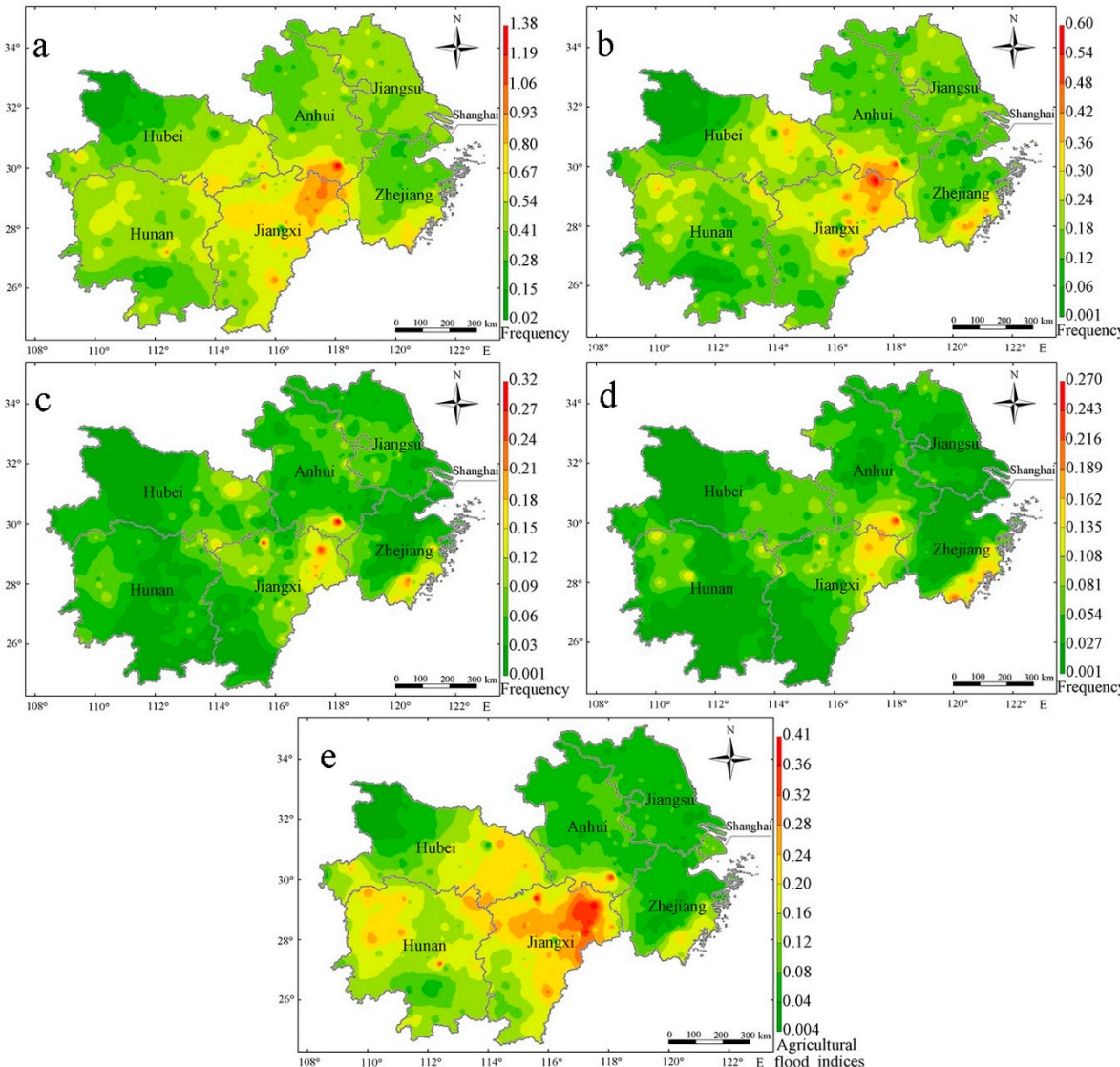

**Figure 5.** Average distribution of each grade of flood frequency: (**a**) Waterlogging; (**b**) medium flooding; (**c**) severe flooding (**d**); catastrophic flooding; and (**e**) agricultural flood indices in middle and lower reaches of Yangtze River from 1970 to 2018. The greater the value of the agricultural flood indices, the higher the probability of agricultural flooding in the region.

The annual agricultural flood indices in the study area were graded according to natural breaks, and the proportion of these indices in each grade was counted (Figure 6).The trend line slopes of low grade, lower grade, medium grade, higher grade, and high grade were −0.15, 0.04, 0.08, 0.04, and 0.04, respectively. Overall, the areas with low agricultural flood indices showed a decreasing trend, whereas those with higher indices showed an increasing trend. In 1994, 1996, 1998, and 2016, the regional areas with high agricultural flood indices were larger than that of other years.

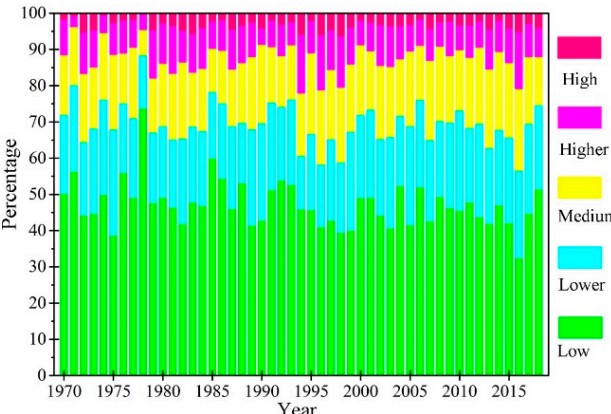

**Figure 6.** Area of each grade of agricultural flood indices in middle and lower reaches of the Yangtze River from 1970 to 2018.

4.2.2. Analysis of Spatial Dynamic Changes in Agricultural Floods

To analyze the distribution and movement trajectory of the agricultural flood center of gravity in the middle and lower reaches of the Yangtze River over the past 49 years, the latitude and longitude of the agricultural flood center from 1970 to 2018 were calculated using the agricultural flood indices as the weight. The distribution map of the agricultural flood gravity center and that of the movement trajectory over the study period are shown in Figure 7. The movement of the agricultural floods center fluctuated, but it was not randomly distributed. Over the past 49 years, the agricultural flooding center of gravity in the study area moved 1570 km in total. The minimum moving speed was 15 km/year from 2000 to 2005, and the maximum was 67 km/year from 1980 to 1985. Affected by precipitation intensity and frequency, the agricultural flooding center of gravity fluctuated greatly with the year. For example, from 1970 to 1975, the center of gravity moved from Jiangxi to Anhui Province, and the agricultural flood frequency in the latter increased by 104.25% during the same period. From 2005 to 2010, the center of gravity was transferred from Anhui to Jiangxi Province, and the frequency of agricultural floods in the latter increased by 117.78%. The agricultural flooding center of gravity in the study area generally alternated between southwest and northeast, and the transfer distance could be divided into three main stages, as follows: It gradually increased in 1970–1985, decreased between 1985 and 2005, and first increased and then decreased during 2005–2018. The main reasons for this may include the fluctuation in the agricultural flood frequency during different periods. For example, during the period of 1970–1985 and 2005–2018, the frequency of agricultural floods changed relatively substantially, while the frequency of agricultural floods was relatively stable during 1985–2005. This may also be related to the spatial extent of agricultural floods (Table 3).

**Table 3.** Moving distance, speed, and direction of the gravity center of agricultural flood indices in middle and lower reaches of Yangtze River.

| Time | Gravity Center of Agricultural Flood Index | | |
|------|-----------|-----------|-----------|
| | Distance (km) | Speed (km/year) | Direction (°) |
| 1970–1975 | 133 | 27 | Northeast 47 |
| 1975–1980 | 258 | 53 | Southwest 75 |
| 1980–1985 | 336 | 67 | Southeast 4 |
| 1985–1990 | 103 | 21 | Northwest 75 |
| 1990–1995 | 98 | 20 | Southwest 55 |
| 1995–2000 | 97 | 19 | Northeast 25 |
| 2000–2005 | 76 | 15 | Northeast 25 |
| 2005–2010 | 190 | 38 | Southwest 48 |
| 2010–2015 | 230 | 46 | Northeast 53 |
| 2015–2018 | 48 | 16 | Northwest 63 |

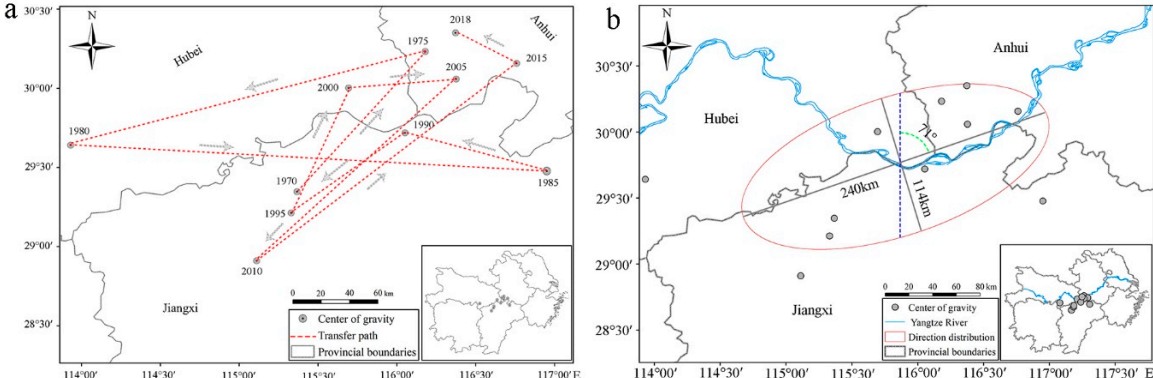

**Figure 7.** (**a**) The movement and (**b**) distribution of the gravity center of agricultural floods in the middle and lower reaches of the Yangtze River from 1970 to 2018.

To further analyze the distribution degree and direction of the agricultural flood gravity center, a standard deviation ellipse of 11 agricultural floods was created using the standard deviation ellipse tool of ArcGIS (Version,10.2; Manufacturer, Environment System Research Institute, ESRI, USA) [42]. The direction of the standard deviation ellipse was 71° northeast, indicating that the agricultural flood center of gravity was generally distributed in a southwest–northeast direction. The standard ellipse long axis was in a southwest–northeast direction, with a length of 240 km, whereas the short axis was in a northwest–southeast direction with a length of 114 km. The ratio of the long to the short axis was about 2.2, indicating that the southwest–northeast dispersion of the agricultural flood center of gravity was greater than northwest-southeast dispersion. Analyses of agricultural floods over the study period showed that agricultural floods frequently occurred in the southwest–northeast direction and particularly at the junction of Anhui, Hubei, and Jiangxi Provinces. However, in the northwest–southeast direction, agricultural flooding only occurred frequently in southeast Hubei Province and southeast Jiangxi Province. The leverage effect was not obvious, which led to a southwest–northeast distribution of the agricultural flooding center of gravity. In addition, the elliptical direction generated by the center of gravity of the 11 studied agricultural floods was similar to that of the Yangtze River. This shows that the agricultural floods were closely related to this river. Watershed characteristics, such as the area, slope, and length of the river basin, constitute the hydrogeographical factors of the area, i.e., the study area contains many plains, the curved river channel, and the slow water flow. With continuous precipitation, the water level can rise rapidly, often causing serious flooding and bringing about huge losses in agricultural production.

### 4.2.3. Characteristic Analysis of Agricultural Flood Spatial Trend

Figure 8 shows the distribution of the propensity rate of the agricultural flood indices in the middle and lower reaches of the Yangtze River from 1970 to 2018. The agricultural flood indices in the study area generally showed an increasing trend, with an average of 0.11%/year. According to the degree of change in the agricultural flood indices, areas with highly significantly reduced, significantly reduced, non-significantly reduced, non-significantly increased, significantly increased, and highly significantly increased agricultural flood indices comprised 0.45, 0.42, 14.13, 73.00, 7.61, and 4.39% of the total area, respectively. Regions with increases in agricultural flood indices were mainly the eastern coastal areas, the junction between Hubei and Hunan Provinces, and northeastern Jiangxi Province. More than half of the study area showed increased agricultural flood indices. Areas with reduced agricultural flood indices were mainly concentrated in the northern part of the study area.

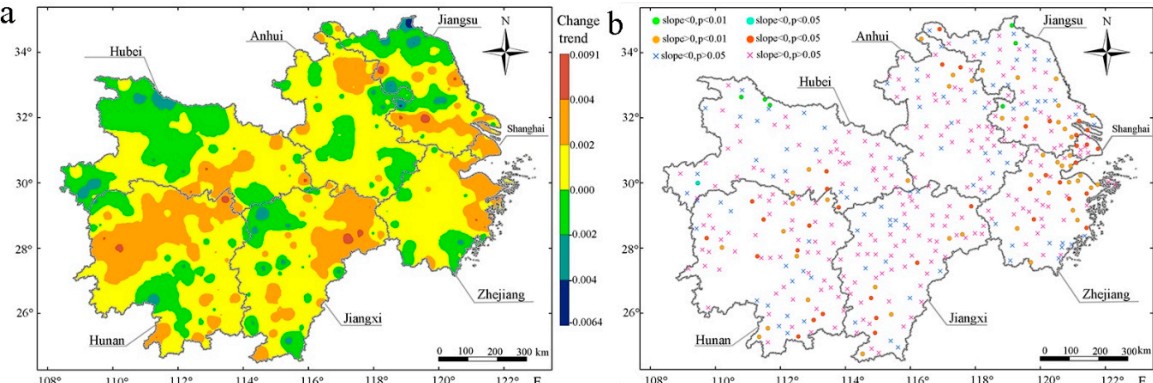

**Figure 8.** (**a**) Climate tendency rate of flood indices and (**b**) significance detection in the middle and lower reaches of the Yangtze River from 1970 to 2018. Colored dots indicate the stations that passed 95% or 99% significance tests. × indicates stations that did not pass a 95% significance test.

## 5. Discussion

### 5.1. Verification of Indicator Rationality and Comparison of Results

This study analyzed the correlation between the average agricultural flood indices and the average annual agricultural flood disaster rate in each Chinese province to obtain the single-station agricultural flood indices (Tables 3 and 4) in the middle and lower reaches of the Yangtze River. The correlation coefficient was tested according to a significance level of $\alpha = 0.01$. The results of this study were compared with relevant literature on flood analyses in 1991, 1996, 1998, and 2016 in the study area to determine whether the selection of agricultural flood indicators was reasonable [43,44]. The aforementioned years were years with frequent flooding. Significant differences existed between severely affected areas in the study area during these years. Heavy flooding occurred in the Jianghuai region during 1991 (Figure 9a), with severe damage concentrated in Hubei, Anhui, and Jiangsu Provinces. Typical rainstorms occurred during 1996 (Figure 9b) in the Meiyu period, with disasters mainly occurring in the middle reaches of the Yangtze River, and with Hunan Province being the most seriously affected. Hubei, Jiangxi, and Anhui Provinces also experienced different levels of flood disasters. In 1998 (Figure 9c), floods occurred in the Yangtze River, and severely damaged areas were mainly concentrated in Hubei, Hunan, Anhui, and Jiangxi Provinces. Flood disasters at the junctions of Hubei, Anhui, and Jiangxi Provinces, Poyang and Dongting lakes, and the middle and lower reaches of the Hanjiang Plain were the most serious. In 2016 (Figure 9d), regional floods occurred in the study area. Hubei, Anhui, Jiangsu, Jiangxi, and Hunan Provinces experienced serious damage, especially at the junction of Hubei, Anhui, and Jiangxi Provinces. According to the flood data of the Yangtze River yearbook, Qin et al. [43] calculated the main flood-affected areas and disaster situations during 1870, 1931, 1935, 1954, 1991, 1995, 1996, 1998, 1999, and 2002. These authors pointed out that

flood disasters were mainly concentrated in the Taihu Basin and parts of Hubei Province during 1991. In 1996, floods mainly occurred in the middle reaches of the Yangtze River Basin. Flood disasters in Hunan Province were the most severe, as was the situation in northeastern Hubei Province and the middle and lower reaches of the Hanjiang River. Severely damaged areas in 1998 included Poyang, Dongting, and Dongshui Lakes, Lishui City, the Jingjiang River section of the Yangtze River, and the middle and lower reaches of the Jianghan Plain. Yu et al. [44] pointed out that the flooding in 2016 was similar to that in 1996, with regional flooding occurring in the middle and lower reaches of the Yangtze River. Hubei, Hunan, Anhui, Jiangxi, and Jiangsu Provinces were seriously affected. These results are essentially consistent with those of this study.

**Table 4.** Single-station flood indicators and agricultural flood indices in the middle and lower reaches of the Yangtze River.

| Province | Annual Agricultural Flood Indices | Correlation Analysis |
|---|---|---|
| Anhui | $P = (0.0320 \times F1 + 0.1684 \times F2 + 0.5947 \times F3 + 0.2049 \times F4)/65$ | 0.8801 |
| Hunan | $P = (0.2906 \times F1 + 0.0665 \times F2 + 0.2397 \times F3 + 0.4032 \times F4)/83$ | 0.7947 |
| Hubei | $P = (0.2016 \times F1 + 0.2756 \times F2 + 0.2471 \times F3 + 0.2757 \times F4)/67$ | 0.7298 |
| Jiangsu | $P = (0.0039 \times F1 + 0.0724 \times F2 + 0.7916 \times F3 + 0.1756 \times F4)/66$ | 0.7628 |
| Jiangxi | $P = (0.2641 \times F1 + 0.0451 \times F2 + 0.0790 \times F3 + 0.6118 \times F4)/83$ | 0.7893 |
| Shanghai | $P = (0.1176 \times F1 + 0.4162 \times F2 + 0.1642 \times F3 + 0.3020 \times F4)/10$ | 0.7478 |
| Zhejiang | $P = (0.0261 \times F1 + 0.2142 \times F2 + 0.4469 \times F3 + 0.3128 \times F4)/64$ | 0.5849 |

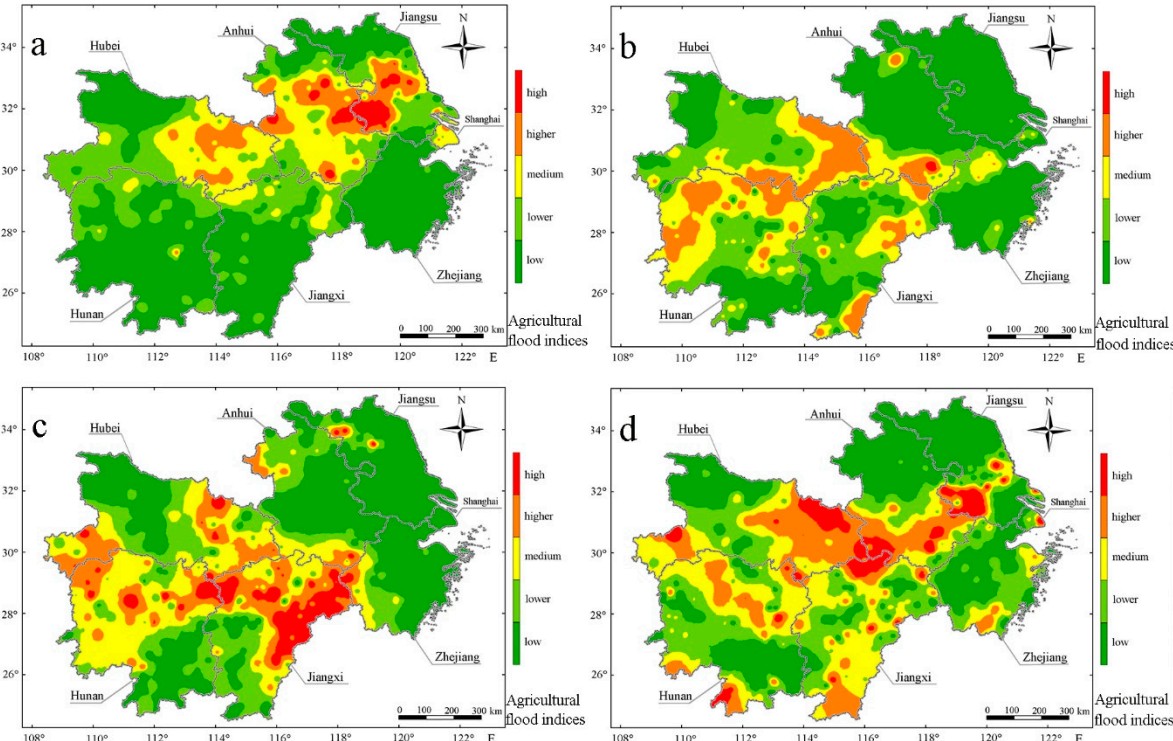

**Figure 9.** Distribution of agricultural flood indices in the middle and lower reaches of the Yangtze River in (**a**) 1991, (**b**) 1996, (**c**) 1998, and (**d**) 2016.

Wang [45] analyzed the spatiotemporal variation characteristics of extreme precipitation in the middle and lower reaches of the Yangtze River over the past 52 years and indicated that the extreme precipitation had annual fluctuations, increasing significantly after 1990. Upon studying the precipitation characteristics of the study area over the past 50 years, He [46] found that the region generally showed an increasing precipitation trend.

*5.2. Limitations of This Study and Directions for Future Research*

The agricultural flood indicators proposed in this study have several limitations, as follows: (1) Rainfall was mainly considered in the construction of the agricultural flood indicators. However, the formation of agricultural floods, including factors such as topography and water conservancy facilities, should also be considered. More detailed data on agricultural flood disasters in the middle and lower reaches of the Yangtze River should be gathered to build smaller-scale regional agricultural indicators. (2) Most of the current agricultural flood indicators are based on mathematical methods for determining the threshold of different precipitation levels, but there is a lack of research on the mechanism of agricultural floods. Research focused on specific crop floods only analyzes the effects of the flooding days and depth on crop growth and yield through flooding tests. Combining the flooding days and depth with precipitation factors to construct agricultural flood indicators is likely to be a fruitful direction for future research. Agricultural flooding could be affected by the spatial distribution of rainfall and by the discharge of large rivers; areas that had not received rainfall could experience serious flooding if the river brought high volumes of water from further upstream. The evolution of agricultural floods under the combined effects of multiple factors will be the focus of further research.

In this paper, fractal analysis can reveal the inherent regularity of the process of time series variation. As the sample increases, making predictions about the future trends of agricultural flood frequency in the region will be valuable. The formation and evolution of floods are caused by a combination of factors. Climate change is considered to be a main cause of flood patterns, intensity, and grade changes [47,48]. Global climate change characterized by increasing temperatures has been recognized for the past century. It is believed that the increase in recent heavy precipitation events is related to global warming [49,50]. In addition, human activities can not only cause extreme temperature fluctuations, but increased greenhouse and ozone-depleting gas emissions are also likely to cause extreme fluctuations in precipitation [51]. Correlation analyses among the average annual temperature, population density, and agricultural flood indices, as Figure 10 shown, indicated that the average correlation coefficient between annual average temperature and agricultural flood indices is 0.0382, and that between population density and agricultural flood indices is 0.0401. This suggests that climate change and human activities have a relationship with the formation and evolution of agricultural floods.

The two dominant factors of climate change and human activities constitute the driving force of the development of floods. Therefore, when studying the spatiotemporal distribution characteristics of agricultural flood disasters in the middle and lower reaches of the Yangtze River, the following should be considered: The special geographical location and complex climatic conditions of this area, including atmospheric circulation anomalies, sea surface temperature anomalies, and thermal and dynamic effects of the Qinghai–Tibet Plateau, affect the precipitation distribution and intensity. Lee et al. [52] pointed out that the probability of extreme summer precipitation in eastern China is related to intraseasonal oscillation in the Asian monsoon region, and strong intraseasonal oscillation events have a significant regulatory effect on the probability of extreme precipitation in eastern China.

As the world's strongest interaction between the ocean and atmosphere, the El Niño–Southern Oscillation causes regional and even global atmospheric circulation anomalies and is a main factor affecting precipitation in China [53]. The thermal and dynamic effects of the Qinghai–Tibet Plateau have a crucial impact on the weather and climate and play a pivotal role in the formation of Asian and even global climate patterns [54]. In addition, local topography has various dynamic and thermal effects on atmospheric motion, i.e., the terrain affects the dynamics, thermal, and microphysical effects of precipitation, which are also the main factors in abnormal local weather, including extreme precipitation [55]. Extreme climate formation and its impacts cannot only be explained by natural fluctuations and natural factors; human activities also play a crucial role. Precipitation models that consider human activities would therefore be useful. It is valuable to understand those impact factors, and use statistical or dynamic models to analyze the nonlinear effects of multi-factor coupling.

Numerical models can also be used to predict the response of agricultural floods to those factors and to identify long-term variations.

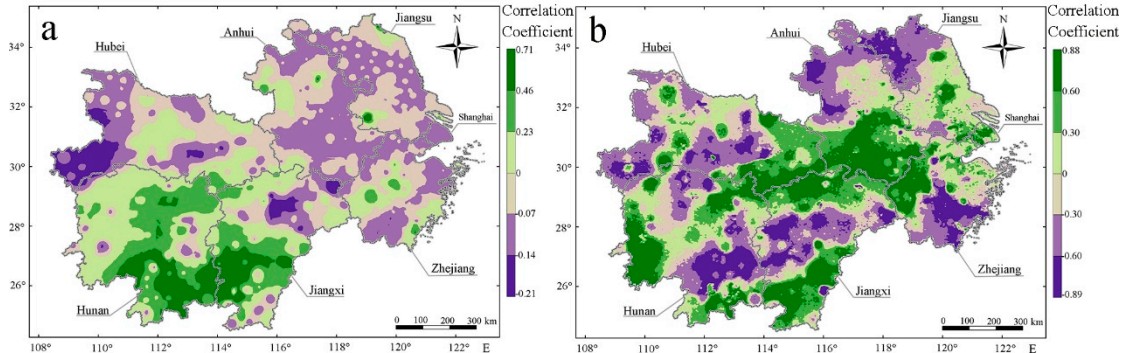

**Figure 10.** (**a**) Correlation coefficients between agricultural flood indices and annual mean temperature; (**b**) Correlation coefficients between agricultural flood indices and population density.

## 6. Conclusions

The study in this paper analyzed, in detail, the spatiotemporal variation characteristics of agricultural floods in the middle and lower reaches of the Yangtze River. This paper provides a scientific basis for regional flood control and disaster mitigation. Precipitation data from 435 meteorological stations, historical disaster records of seven provinces and cities, and agricultural flood areas of the middle and lower reaches of the Yangtze River were studied. Mann–Kendall tests, wavelet analyses, trend analyses, and center of gravity modeling were used to study the spatiotemporal changes and development trends in agricultural floods in the middle and lower reaches of the Yangtze River based on agricultural flood indicators and rational verification of these indicators. The following conclusions were drawn, as follows:

(1) Trend and anomaly analyses of the frequency of agricultural floods in the study area were conducted. Changes in the frequency of agricultural floods were phased and showed a generally increasing trend. The area experienced "growth–reduction–growth–reduction–growth" fluctuations. Based on Mann–Kendall abrupt change detection and cumulative anomaly analyses, the frequency of agricultural flooding abruptly changed in 1990. Subsequently, the average frequency of agricultural floods per station changed from a decreasing to an increasing trend. The average frequency of agricultural floods per station increased by 0.2086/year in 1991–2018.

(2) The agricultural flood frequency of the study area was analyzed using Morlet wavelets. The frequency of agricultural floods had periodic variations on three time scales, as follows: 2–4, 5–10, and 13–30 years. The 17-year time scale had three cycles from low to high and constituted the first major period of the annual frequency of agricultural flooding. The 8-year time scale, which had six cycles, was the second major period, and the 4-year time scale was the third major period, with 13 cycles.

(3) Analyses of the distribution characteristics of agricultural floods in the study area showed that agricultural floods were concentrated in the southern Yangtze River, and fewer floods occurred in the north. These frequently flooded areas were mainly located in low-lying areas and around lakes and rivers, which are prone to flooding. The annual agricultural flood indices were graded, and the area ratio of these indices at each level was counted. The results showed that areas with low agricultural flood indices showed a decreasing trend, whereas those with higher agricultural flood indices showed an increasing trend.

(4) The gravity center of agricultural floods was affected by the intensity and frequency of precipitation. It fluctuated greatly over the years and generally alternated from the southeast to northwest. The dispersion of the gravity center in a southwest–northeast direction was greater than that in a northwest–southeast direction. The direction of agricultural flooding was similar to that of the Yangtze River, indicating that agricultural floods in the study area are closely related to this river.

(5) Slope trend analyses revealed that the agricultural flood indices in the study area generally showed an increasing trend, with an average of 0.11%/year. The areas with highly significant reductions, significant reductions, non-significant reductions, non-significant increases, significant increase, and highly significant increases in agricultural flood indices represented 0.45, 0.42, 14.13, 73.00, 7.61, and 4.39% of the total area, respectively.

**Author Contributions:** Conceptualization, S.W., W.Z. and S.C.; Data curation, S.W., S.C.; Formal analysis, S.W. and W.Z.; Funding acquisition, W.Z.; Investigation, Z.W. and S.L.; Methodology, S.W.; Project administration, W.Z.; Resources, X.Q.; Supervision, W.Z.; Validation, S.W.; Visualization, S.W. Writing-original draft, S.W.; Writing-review & editing, S.W.

**Funding:** This study was supported by the National Key R&D Program of China (no. 2017YFC1502901).

**Acknowledgments:** We acknowledge the use of data from China Meteorological Administration.

**Conflicts of Interest:** The authors declare no conflict of interest.

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
