# Peer review of "Spatiotemporal Variations in Agricultural Flooding in Middle and Lower Reaches of Yangtze River from 1970 to 2018"

_sustainability, doi:10.3390/su11236613_

Round 1
Reviewer 1 Report
Comments and Suggestions for Authors
This paper analyses spatiotemporal changes in floods frequency and agricultural flood indices using trend analyses, wavelet analyses, and gravity center, etc. The results show the flood characteristics (trend and periodic oscillation), the proportion of area with different flood occurrence and gravity center movement in Yangtze River from 1970 to 2018. It provided the comprehensive perspectives about the variation of climate pattern with time.
The content and each section are distinct and well organized. Only content narratives need to avoid redundant words for readers to understand easily. The following are some comments about the paper content and the method:
Page 3, line 114 to page 4, line 116: Please reword “Precipitation process” and “daily precipitation less than 0.1 mm”.
Such as line 146, 148, 158, etc., the format (font size and alignment) of the equations or variables in the several sentences needs to be adjusted. Please check and correct.
Page 9, line 247-248: The duration explored in “Three declines occurred in the overall increasing trend, namely in 1975-1980, 1984-1989, and 2000-2005.” can’t correspond to the Figure 2(b). Please check and correct.
In section 4.1.2, this study mentioned “the high-low variation of the large time scale of the annual flood frequency included the high-low variation of the smaller time scale.”, is there any explanation for this situation or phenomenon? Please describe it in the content.
Page 10, line 296-301: For the discussion of areas with flood occurrence, several region/river/lakes are mentioned in the content. However, the location didn’t show in Figure 1 or 5, causing confusion in understanding. Please check and revise.
Page 11, line 314-317: The flood indices was graded to understand the proportion of area for different flood level, but why grades are showed as equations instead simple values, and what the equations mean?
Page 11, line 318: The areas with high flood indices “in 1995” is relatively smaller in Figure 6. Please check and correct.
Page 12, line 336-338: The study mentioned “The agricultural flooding center of gravity in the study area generally alternated between southeast and northwest” here, but the following results indicate that the distribution is southwest-northeast. Is there a contradiction? Please check and correct.
Page 12, line 338-340: “the transfer distance could be divided into three main stages: It gradually increased in 1970-1985, decreased between 1985 and 2005, and first increased and then decreased during 2005- 2018.” How these three stages are divided? and why? (alternate significantly or other reasons?) Please explain and describe about it in the content.
Page 13, line 360-361: Please reword the sentence “the study area contains many plains, the river channel is curved, and the water flow is slow” rewords as “the study area contains many plains, the curved river channel, and the slow water flow.”
Please sum up the contribution of this study in Conclusion.
Page 15, line 423: Please add the reference about “relevant literature on flood analysis.”
Page 15-17: Suggest Discussion can be placed before Conclusion.
In References: Please check all the format is consistent and same as journal rules, such as year (bold font) in [3], extra () in [37], dot before year, etc.
Author Response
28 Oct 2019
Ms. Ref. No.: sustainability-612782
Title: Spatiotemporal variations in agricultural flooding in middle and lower reaches of Yangtze River from 1970 to 2018
Dear Reviewer:
Thank you very much for giving us comments concerning our manuscript. Those insightful comments were valuable and very helpful for revising and improving our paper, and they have provided important guidance to our research. We have studied the comments carefully and have made corrections that we hope will be met with your approval. We have used “Track Changes” function in Microsoft Word so that it can be clearly identified. In addition, we employed an English-language editing service, Editage, to polish our wording. Certification is attached. We believe that this paper is a significant improvement over the last version. The main corrections in this paper are as follows:
This paper analyses spatiotemporal changes in floods frequency and agricultural flood indices using trend analyses, wavelet analyses, and gravity center, etc. The results show the flood characteristics (trend and periodic oscillation), the proportion of area with different flood occurrence and gravity center movement in Yangtze River from 1970 to 2018. It provided the comprehensive perspectives about the variation of climate pattern with time.
The content and each section are distinct and well organized. Only content narratives need to avoid redundant words for readers to understand easily. The following are some comments about the paper content and the method:
Page 3, line 114 to page 4, line 116: Please reword “Precipitation process” and “daily precipitation less than 0.1 mm”.
Response: We have revised “Process precipitation refers to continuous precipitation of more than 50 mm over 1 day. When daily precipitation of less than 0.1 mm occurs over multiple days, the process of cumulative precipitation is considered interrupted” to “The process of cumulative precipitation refers to continuous precipitation of more than 50 mm over 1 day. When less than 0.1 mm of daily precipitation occurs over multiple days, the process of cumulative precipitation is considered interrupted”. (Page 4, line 125 to 128)
Such as line 146, 148, 158, etc., the format (font size and alignment) of the equations or variables in the several sentences needs to be adjusted. Please check and correct.
Response: We have adjusted the format of the equations or variables in line 157, 159, 173, 176, 179-184, 197-199, 215, 235.
Page 9, line 247-248: The duration explored in “Three declines occurred in the overall increasing trend, namely in 1975-1980, 1984-1989, and 2000-2005.” can’t correspond to the Figure 2(b). Please check and correct.
Response: We have checked and corrected this part of the article:” Three declines occurred in the overall increasing trend, namely in 1974–1980, 1982–1986, and 1997–2001.”(Page 9, line 265,266)
In section 4.1.2, this study mentioned “the high-low variation of the large time scale of the annual flood frequency included the high-low variation of the smaller time scale.” is there any explanation for this situation or phenomenon? Please describe it in the content.
Response: We just mentioned the sentence but not give any explanation for this phenomenon. We have revised as:” In addition, the high–low variation of the large time scale of the annual flood frequency included the high–low variation of the smaller time scale, for example, the higher period of the first agricultural flood frequency at the 17-year scale included one high and one low period on the 8-year time scale and two low and one high periods on the 4-year time scale, and lower period of the second agricultural flood frequency at the 17-year scale included one high and one low period on the 8-year time scale and two low and two high periods on the 4-year time scale”(Page 10, line 298 to 303)
Page 10, line 296-301: For the discussion of areas with flood occurrence, several region/river/lakes are mentioned in the content. However, the location didn’t show in Figure 1 or 5, causing confusion in understanding. Please check and revise.
Response: We have checked this part in the paper, and revised the Fig.1.c. We have added those data in it. (Page 3, Fig. 1.c)
Page 11, line 314-317: The flood indices was graded to understand the proportion of area for different flood level, but why grades are showed as equations instead simple values, and what the equations mean?
Response: Thank you very much for your valuable suggestion, we may not have used the formula reasonably. Using simple values does make the readers easier to understand, and make the paper more concise. We have revised as:” The trend line slopes of low grade, lower grade, medium grade, higher grade and high grade were -0.15, 0.04, 0.08, 0.04, and 0.04, respectively.”(Page 11, line 335 to 338)
Page 11, line 318: The areas with high flood indices “in 1995” is relatively smaller in Figure 6. Please check and correct.
Response: Thank you very much for your review, we are so sorry that this error has appeared in the paper. We have revised “in 1995” to “in 1994” after my check. (Page 11, line 340)
Page 12, line 336-338: The study mentioned “The agricultural flooding center of gravity in the study area generally alternated between southeast and northwest” here, but the following results indicate that the distribution is southwest-northeast. Is there a contradiction? Please check and correct.
Response: We have check this part in the paper, and the description does not match the results. We have revised in the paper:” The agricultural flooding center of gravity in the study area generally alternated between southwest and northeast” (Page 12, line 359, 360) and we have also made corresponding revise in the abstract. (Page 1, line 33)
Page 12, line 338-340: “the transfer distance could be divided into three main stages: It gradually increased in 1970-1985, decreased between 1985 and 2005, and first increased and then decreased during 2005- 2018.” How these three stages are divided? And why? (Alternate significantly or other reasons?) Please explain and describe about it in the content.
Response: In the original manuscript, we just described the phenomenon but not give a reasonable explanation. We have revised as:” It gradually increased in 1970–1985, decreased between 1985 and 2005, and first increased and then decreased during 2005–2018. The main reasons may include the fluctuation of agricultural flood frequency in different periods. For example, during the period of 1970-1985 and 2005-2018, the frequency of agricultural floods changed relatively great, while during 1985-2005, the frequency of agricultural floods was relatively stable. It may also be related to the spatial extend of agricultural floods.”(Page 12, line 362 to 366)
Page 13, line 360-361: Please reword the sentence “the study area contains many plains, the river channel is curved, and the water flow is slow” rewords as “the study area contains many plains, the curved river channel, and the slow water flow.”
Response: We have reworded as “the study area contains many plains, the curved river channel, and the slow water flow.”(Page 13, line 386, 387)
Please sum up the contribution of this study in Conclusion.
Response: We have summed up the contribution of this study in Conclusion as” This paper analyzed spatiotemporal variation characteristics of agricultural floods in detail in the middle and lower reaches of the Yangtze River, and provided a scientific basis for regional flood control and disaster mitigation. ” (Page 17, line 542 to 544)
Page 15, line 423: Please add the reference about “relevant literature on flood analysis.”
Response: We have added the reference” The results of this study were compared with relevant literature on flood analyses in 1991, 1996, 1998, and 2016 in the study area to determine whether the selection of agricultural flood indicators was reasonable[43,44]”.(Page 15, line 449 to 451)
Page 15-17: Suggest Discussion can be placed before Conclusion.
Response: We have placed suggest discussion before conclusion.
In References: Please check all the format is consistent and same as journal rules, such as year (bold font) in [3], extra () in [37], dot before year, etc.
Response: We have checked all the format in reference (line 593, 594, 595, 598, 599, 601, 610, 612, 618, 627, 641, 651 to 654, 655, 660, 664, 676, 678 to 682, and 686)
We deeply appreciate your thoughtful and professional work and hope that the corrections will be met with approval.

Reviewer 2 Report
Among the statistical methods, it would be interesting to try fractal analysis. Perhaps this should be mentioned among areas of further research.
Author Response
28 Oct 2019
Ms. Ref. No.: sustainability-612782
Title: Spatiotemporal variations in agricultural flooding in middle and lower reaches of Yangtze River from 1970 to 2018
Dear Reviewer:
Thank you very much for your comments concerning our manuscript. Those insightful comments were valuable and very helpful for revising and improving our paper, and they have provided important guidance to our research. We have studied your comments carefully and have made corrections that we hope will be met with your approval. We have used “Track Changes” function in Microsoft Word so that it can be clearly identified. In addition, we employed an English-language editing service, Editage, to polish our wording. Certification is attached. We believe that this paper is a significant improvement over the last version. The main corrections in this paper are as follows:
Among the statistical methods, it would be interesting to try fractal analysis. Perhaps this should be mentioned among areas of further research.
Response: Thank you very much for your valuable suggestion, and we have added in the paper:” In the research of this paper, fractal analysis can reveal the inherent regularity of the process of time series variation. As the sample increases, making predictions about the future trends of agricultural flood frequency in the region will be valuable.”(Page 16, line 501 to 503)
We tried our best to improve the manuscript and made major revisions in the manuscript, as well as the English language. We deeply appreciate the your thoughtful and professional work and hope that the corrections will be met with approval.

Reviewer 3 Report
The paper deals with a highly important topic that requires much scientific input. The idea of looking at changes and trends in the flooding of agricultural land in the middle and lower Yangtze basin is excellent. The data and methodology of the paper needs to be explained more carefully. The actual statistics used in your analysis could be set out in a table. It seems that you used a form of categorical data rather than linear data sets. You do not examine how this affects the validity of the results you obtained.
The idea of looking at the centre of gravity of agricultural flooding is strange as it is going to be affected by the spatial distribution of rainfall and by the the discharge (flow) of the great rivers; an areas that had not received rainfall could experience serious flooding if the river brought high volumes of water from further upstream. You do not seem to deal with this issue.
The presentation of results with four decimal places is unusual in hydrological work as the measurements are not to that level of accuracy.
There is no real discussion of how climate change affects the potential use of your methods: some of your graphs suggest that since 1990 rainfall volumes have increased. Certainly in most parts of the world storminess is increasing, which affects flooding magnitudes.
Such themes need careful attention in a revision of the paper. I have annotated the attached pdf of the paper.

Author Response
28 Oct 2019
Ms. Ref. No.: sustainability-612782
Title: Spatiotemporal variations in agricultural flooding in middle and lower reaches of Yangtze River from 1970 to 2018
Dear Reviewer:
Thank you very much for your comments concerning our manuscript. Those insightful comments were valuable and very helpful for revising and improving our paper, and they have provided important guidance to our research. We have studied your comments carefully and have made corrections that we hope will be met with your approval. We have used “Track Changes” function in Microsoft Word so that it can be clearly identified. In addition, we employed an English-language editing service, Editage, to polish our wording. Certification is attached. We believe that this paper is a significant improvement over the last version. The main corrections in this paper are as follows:
The paper deals with a highly important topic that requires much scientific input. The idea of looking at changes and trends in the flooding of agricultural land in the middle and lower Yangtze basin is excellent. The data and methodology of the paper needs to be explained more carefully. The actual statistics used in your analysis could be set out in a table. It seems that you used a form of categorical data rather than linear data sets. You do not examine how this affects the validity of the results you obtained.
The idea of looking at the center of gravity of agricultural flooding is strange as it is going to be affected by the spatial distribution of rainfall and by the discharge (flow) of the great rivers; an areas that had not received rainfall could experience serious flooding if the river brought high volumes of water from further upstream. You do not seem to deal with this issue.
The presentation of results with four decimal places is unusual in hydrological work as the measurements are not to that level of accuracy.
There is no real discussion of how climate change affects the potential use of your methods: some of your graphs suggest that since 1990 rainfall volumes have increased. Certainly in most parts of the world storminess is increasing, which affects flooding magnitudes.
Such themes need careful attention in a revision of the paper. I have annotated the attached pdf of the paper.
The actual statistics used in your analysis could be set out in a table.
Response: We have set out statistics in a table.
Table 1. The statistics used in the paper
|
Data Type |
Data Sources |
|
Planting area(ha) |
Statistical yearbooks |
|
Damage-suffered people(population) |
Statistical yearbooks |
|
Damage area(ha) |
Statistical yearbooks |
|
Economic loss(trillion yuan) |
Meteorological disaster books |
|
Flood depth(m) |
Meteorological disaster books |
|
Flood duration(d) |
Meteorological disaster books |
(Page 3, line 115, 116)
It seems that you used a form of categorical data rather than linear data sets. You do not examine how this affects the validity of the results you obtained.
Response: The precipitation data used in the paper was derived from meteorological stations. The indicators of agricultural floods were selected by continuous rainfall, and the weights of each indicator were determined according to the proportion of inundated area. Verification can determine the rationality of the indicator selection. In this step, we also referred to the relevant literature, and the author verified the rationality of various agricultural flood indicators. [Zhang G.; Huo Z.; Wu L. The temporal and spatial variations of agricultural flood disaster over the middle and lower reaches of the Yangtze River from 1961 to 2010.Geographical Research.2015, 34, 1097-1108. (In Chinese)]. We really hope our opinion will meet with your approval.
Page 1, line 41: insert “its”
Response: We have inserted “its”, as” China uses 7% of its cultivated land to support 22% of its population”. (Page 1, line 41)
Page 1, line 42: whose food requirements? The nation's? The province's? The local city's?
Response: We are so sorry that we haven’t expressed clearly. And we have added “the nation’s” as “Over the next 20 years, the nation’s food requirements are expected to increase by 30–50%”.
Page 2 ,line 43 to 45: Rewrite these sentences: 'The flat terrain of the middle and lower reaches of the Yangtze River, which supports China’s major agricultural production areas, experience a monsoon climate which often brings flooding'.
Response: We have revised “The middle and lower reaches of the Yangtze River are China’s major agricultural production areas, and it has a flat terrain and large population. These areas are located in monsoon climate zones and are often threatened by floods” to “The flat terrain of the middle and lower reaches of the Yangtze River, which supports China’s major agricultural production areas, experiences a monsoon climate that often brings about flooding”. (Page 2, line 43 to 47)
Page 2, line 46: Have you defined what you mean by spatiotemporal? Increased variations of rainfall over space and time?
Response: We are so sorry that we haven’t defined spatiotemporal and we have revised “spatiotemporal changes in precipitation” as “Increased spatiotemporal variations of rainfall”. (Page 2, line 48)
Page 2, line 46: Concentration of what? Economic activity? Land values? Rich people?
Response: Probably because the translation is not very suitable, we haven’t express clearly and we have revised “high economic concentration” to “concentrated areas of high economic activity”. (Page 2, line 49)
Page 2, line 49: Understanding of...is necessary for good scientific understanding in irdwr to achieve regional flood control...
Response: We have revised” Studying the spatiotemporal variation characteristics of agricultural flood disasters in the middle and lower reaches of the Yangtze River is important for providing a scientific basis for regional flood control and disaster mitigation.” as “Understanding the spatiotemporal variation characteristics of agricultural flood disasters is necessary for good scientific understanding in the middle and lower reaches of the Yangtze River to achieve regional flood control and disaster mitigation.’’ (Page 2, line 52 to 57)
Page 2, line 54: maybe it is not just meteorological STATION data but other data such as satellite observations and weather radar. Deleting "station" makes the sentence easier to follow.
Response: We have deleted “station”. (Page 2, line 60)
Page 2, line 57: Revise “using” to “the use of”
Response: We have revised “using” to “the use of’’. Page 2, line 63)
Page 2, line 64: “Normalized” Is this the best word to use? : What did this author do with the normalized data?
Response: We have re-read the quoted article and it has mentioned “The normalization method is used to remove the inconsistency of the disaster results caused by regional differences.” “Normalized” is not suitable so I have revised “normalized” to “used”. (Page 2, line 71
Page 2, line 72 to 73: You already mentioned this in line 38
Response: We have mentioned “Extreme weather events and floods have occurred more frequently since the 1960s, bringing about various environmental problems and causing huge economic losses and severe challenges to sustainable development” in line 38, so I have deleted this sentence “Agricultural floods in the middle and lower reaches of the Yangtze River over the past 50 years have increased both in area and frequency [30, 31]”.Also the two papers were deleted from the references. (Page 2, line 78 79
Page 2, line 73 to 74: Why "therefore" when next you talk about rainstorms?
Why "prototype indicators: what exactly are they?
Response: We have used inappropriate words in the process of translating Chinese into English, which made it difficult to understand. And we have deleted this sentence because I founded that rewriting the next sentence can be better expressed. We have revised to “In this paper, we used meteorological data combined with actual agricultural flood data to construct indicators…” (Page 2, line 78 to 80)
Page 2, line 75 to 77: It isn't easy to understand why are you talking about "real-time" when you have emphasized past floods. Real-time means when floods are actually happening...predicting from incoming rain data when the river is going to peak at a given place.
Response: We are so sorry that we have used the word incorrectly, and we have revised to “In this paper, we used meteorological data combined with actual agricultural flood data to construct indicators, and tested the practicality of indicators in business of agricultural flood monitoring and early warning assessment.” (Page 2, line 80 to 84)
Page 2, line 82: Comprise is not the correct word: contain might be better, because the lakes are not the only features of the plain: there is the agricultural land as well.
Response: Thank you very much for pointing out if the use of the words were reasonable, and we have revised “comprise” to “contain’’. (Page 2, line 88)
Page 2, line 85: Summer is a season, not a year so that phrase needs attention. How large is large? Can you give figures for one or two stations to show this variation over recent decades?
Response: Thank you very much for your valuable suggestion. We have delete “especially in summer” and the sentence revised to “The area has a north subtropical monsoon climate, with more than 50% of the precipitation occurring in June–August, and the annual variation in precipitation is large”. In addition, we have given a figure for Yunxi Station to show the variation over recent decades.
Page 2, line 88: What is a flood: when water comes across the road or when a given area is inundated: What was the definition of "flood" used to count these 1092 floods?: probably they were the huge extensive floods all along the river: but as you have mentioned localized rainstorm flooding, you need to be precise here.
Response: The subject of this discussion was rainstorm floods. We have revised “floods” to “rainstorm floods”. (Page 2, line 94)
Page 3, line 101: Is this data on the spatial extent of flooding and flood losses?
Response: This data was on the spatial extend of flooding and flood losses, and we have revised the sentence to “Agricultural flood data were derived from the statistical yearbooks of the provinces and cities in the study, including the spatial extend of flooding, damaged and inundated areas.” (Page 3, line 108, 109)
Page 3, line 104: Is this information on flood depth and duration? Make sure your expressions will be fully understood by international readers.
Response: We are so sorry that we haven’t expressed clearly, and we have revised the sentence to “Historical flood data, including the flood depth, duration, and disaster losses during flood in different provinces from 1970 to 2018, were mainly obtained from meteorological disaster books and records.” (Page 3, line 110 to 112)
Page 3, line 104: It is not clear what a flood disaster rate is? Do you mean frequency?
Response: We are so sorry that we have not expressed clearly and we have revised “flood disaster rate” to “the proportion of inundated area”. (Page 4, line 121)
Page 4, line 122: the three days
Response: We have added “the”. (Page 4, line 133)
Page 5, line 152: These words are not appropriate and a better form of words is required. WE know that the climate is unstable, so we should we think there is stability? Some better explanation is required.
Response: We are so sorry that these words are not appropriate and we have revised “The Mann-Kendall method is based on the premise of climate sequence stability. The sequence is random and independent, and its probability distribution is equivalent” to “For abrupt climate change detection, Mann-Kendall nonparametric detection method has the characteristics of wide detection range, less human interference and a high degree of quantitative. The sample does not have to comply with a certain distribution, and is not subject to interference by a few outlier.” (Page 5, line 163 to 167)
Page 6, line 172: Another phrase the requires explanation: flood frequency implies knowledge of the return period of specific events, but it is difficult to see how you obtained such data.
Response: We have deleted “changes” and revised the sentence to “According to table.1, we got the cumulative frequency of floods. The time series of flood frequency, due to the influence of natural factors, has complex, nonlinear, and multi-time-scale variation characteristics.” (Page 6, line 186, 187)
Page 7, line 207 to 208: The logic behind wanting to do this is unclear. Why do you expect there to be something special about the center of gravity: how does it relate to where rain actually fell and the types of weather systems that produce flooding rains? Is there a difference due to flood coming down the different tributaries of the Yangtze River?
Response: The application fields of gravity center models are mainly divided into two categories: single factor research, two-factor comparison or coupling study [Gao, J.; Xie, W.; Han, Y. The Evolutionary Trend and the Coupling Relation of Gravity Center Moving of County-level Population Distribution, Economi-cal Development and Grain Production During 1990-2013 in Henan Province. Scientia Geographica Sinica, 2018, 38, 919-926. (In Chinese)]. We mainly considered rainstorm flooding in this paper, and studied the distribution of the center of gravity and the relationship with the Yangtze River. We have revised “This study used the gravity center migration model to quantitatively study the spatial variation in agricultural floods in the middle and lower reaches of the Yangtze River” to “The center of gravity and its moving direction, moving distance can not only show the regional geographical phenomenon spatial difference, but also show its dynamic process and evolution law. This study used the gravity center migration model to quantitatively study the spatial variation in agricultural floods in the middle and lower reaches of the Yangtze River”. But we haven’t considered” agricultural flooding is going to be affected by the discharge (flow) of the great rivers”, and we have added” Agricultural flooding could be affected by the spatial distribution of rainfall and by the discharge of the great rivers; an areas that had not received rainfall could experience serious flooding if the river brought high volumes of water from further upstream. The evolution of agricultural floods under the combined effects of multiple factors will be the focus of further research.” in the limitations of the study. (Page 16, line 495 to 499)
Page 7, line 212 to 213::This kind of statement should come earlier: around line 106.
Response: We have moved “This study utilized the middle and lower reaches of the Yangtze River as the main study area, and the counties and cities under its jurisdiction were used as subregions” to line 113. (Page 7, line 113, 114)
Page 9, line 249: What is this UF curve? (Also the UB curve)
Response: We are so sorry that we haven’t express clearly, I have mentioned c1 ( forms a curve.Page 6, line 176) and c2 ( was drawn as curve .Page 6, line 180) and c1 is UF, c2 is UB. The expression before and after didn’t correspond, and I have revised c1 to UF, c2 to UB.(Page 6, line 178 to 183)
Page 11, line 310 to 312: Need to remind the reader what the values in the y-axis actually mean, for example in graph E does 0.41 at the top of the scale mean a 43% chance of having a flood in any given year?
Response: We have added “The greater the value of the agricultural flood indices, the higher the probability of agricultural flooding in the region.”(Page 11, line 330, 331)
Page 13, table 2: You should not claim to be accurate to four places of decimals in this table (and in other data you use) the nearest km would seem to be accurate enough given the spacing of the meteorological stations
Response: We have revised the number of decimal places and we have expressed the distance and speed as integers in the table and in other data we used. (Page 14, table 3) We have also revised “15.22km/year” to”15km/year” and “67.18km/year” to”67km/year”. (Page 13, line 352)
|
Time |
Gravity center of agricultural flood index |
||
|
Distance (km) |
Speed (km/year) |
Direction (°) |
|
|
1970–1975 |
133 |
27 |
North by east 47 |
|
1975–1980 |
258 |
53 |
South by west 75 |
|
1980–1985 |
336 |
67 |
South by east 4 |
|
1985–1990 |
103 |
21 |
North by west 75 |
|
1990–1995 |
98 |
20 |
South by west 55 |
|
1995–2000 |
97 |
19 |
North by east 25 |
|
2000–2005 |
76 |
15 |
North by east 25 |
|
2005–2010 |
190 |
38 |
South by west 48 |
|
2010–2015 |
230 |
46 |
North by east 53 |
|
2015–2018 |
48 |
16 |
North by west 63 |
Page 14, figure 8. b: the meaning of this key is unclear
Response: We have added “Colored dots indicates stations have passed 95% or 99% significant tests. × indicates stations have not passed a 95% significant test.”(Page 14, line 405, 406)
Page 17, line 490 to 499: These remarks are all highly appropriate but the paper has not really considered how they have operated over the study period.
Response: This passage expressed the factors affecting precipitation in the middle and lower reaches of the Yangtze River, but we haven’t consider how they have operated ,and we have added “It is valuable to understand those impact factors, and use statistical or dynamic models to analysis the nonlinear effects of multi-factor coupling. Numerical models can also be used to predict response of agricultural floods to those factors and identify long-term variations.” (Page 17, line 533 to 536)
We tried our best to improve the manuscript and made major revisions in the manuscript, as well as the English language. We deeply appreciate your thoughtful and professional work and hope that the corrections will be met with approval.

Round 2
Reviewer 1 Report
After MAJOR revision author have provided a sufficient amount of new and important information and therefore recommends to accept the paper.
Author Response
Dear reviewer,
Thank you very much for your approval.
Reviewer 3 Report
There are some errors of language and style which I have indicated on the attached pdf
